# Topic-driven toxicity: Exploring the relationship between online toxicity and news topics

Joni Salminen[1,2]*, Sercan Sengün[3], Juan Corporan[4], Soon-gyo Jung[1], Bernard J. Jansen[1]

**1** Qatar Computing Research Institute, Hamad Bin Khalifa University, Doha, Qatar, **2** University of Turku, Turku, Finland, **3** College of Fine Arts, Illinois State University, Normal, Illinois, United States of America, **4** Banco Santa Cruz RD, Santo Domingo, Dominican Republic

* jsalminen@hbku.edu.qa

**Data Availability Statement:** The comments cannot be shared publicly because they are collected from a proprietary YouTube channel and the channel owner has not authorized their sharing. The data underlying the statistical analyses

## Abstract

Hateful commenting, also known as 'toxicity', frequently takes place within news stories in social media. Yet, the relationship between toxicity and news topics is poorly understood. To analyze how news topics relate to the toxicity of user comments, we classify topics of 63,886 online news videos of a large news channel using a neural network and topical tags used by journalists to label content. We score 320,246 user comments from those videos for toxicity and compare how the average toxicity of comments varies by topic. Findings show that topics like Racism, Israel-Palestine, and War & Conflict have more toxicity in the comments, and topics such as Science & Technology, Environment & Weather, and Arts & Culture have less toxic commenting. Qualitative analysis reveals five themes: Graphic videos, Humanistic stories, History and historical facts, Media as a manipulator, and Religion. We also observe cases where a typically more toxic topic becomes non-toxic and where a typically less toxic topic becomes "toxified" when it involves sensitive elements, such as politics and religion. Findings suggest that news comment toxicity can be characterized as *topic-driven toxicity* that targets topics rather than as vindictive toxicity that targets users or groups. Practical implications suggest that humanistic framing of the news story (i.e., reporting stories through real everyday people) can reduce toxicity in the comments of an otherwise toxic topic.

## Introduction

Online toxicity, defined as hateful communication that is likely to cause an individual user leave a discussion [1], can manifest itself in various ways, including cyberbullying [2], trolling [3], and the creation of online firestorms, defined as "rapid discharges of large quantities of negative, often highly emotional posts in the social media environment" [4] (p. 286), where participants attack other groups or organizations. According to Patton et al. [5], online toxicity may result in violent actions also in the physical world and should, therefore, be treated as a matter with serious social gravity.

presented in the study are available within the paper and its Supporting Information files.

**Funding:** JC is a paid employee of Banco Santa Cruz. The funding of the publication of this research will be done by Qatar National Library (QNL). The funders did not have any additional role in the study design, data collection and analysis, decision to publish, or preparation of the manuscript. The specific roles of these authors are articulated in the 'author contributions' section.

**Competing interests:** JC is employed by Banco Santa Cruz. The funding of the publication of this research will be done by Qatar National Library (QNL). This does not alter our adherence to PLOS ONE policies on sharing data and materials.

Online hate speech is can be seen as old as the Internet itself. Anti-Semitic and racist hate groups were active on Bulletin Board Systems as early as 1984 [6]. In the present time, some communities are specifically geared towards promoting hate speech and providing avenues for expressing politically incorrect values that may not comfortably be expressed in face-to-face interactions [7,8]. Toxic commenting has also been found prevalent in general online discussion forums, news websites, and social media platforms. The existing research deals with multiple aspects, such as detection and classification of toxicity [9–11], assessing its impact on online communities [12,13], types of toxicity such as cyberbullying and trolling [2,14], and means of defusing online toxicity [15]. To approach toxicity, researchers have investigated multiple social media platforms, such as Twitter, YouTube, Facebook, and Reddit [7,11], as well as comments in online discussion forums and news websites [16]. Due to its high prevalence, toxicity has been identified as a key concern for the health of online communities.

Additionally, previous research has identified several risks from new technology to news dissemination and journalism, including clickbait journalism [17], fake news [18], manipulation of search rankings and results to alter public opinion [19,20], and "story hijacking", i.e., repurposing the original story [4]. For example, when the New York Police Department (NYPD) invited the community to share positive experiences, the move backfired, and 70 000 tweets of police brutality were shared alongside the hashtag #MyNYPD [4]. Despite the large amount of research focused on these two areas–online toxicity and the negative impact of technology on news–the relationship between *news topics and online toxicity* remains an unexplored research question. Even though prior research suggests an association between news topics and toxic comments, this association has not been empirically established. The previous studies suggest that political topics can cause hateful debates when associated with group polarization [21], i.e., a strong division to opposing groups among online users. In their study, Zhang et al. [22] considered topic as a feature in machine learning but did not provide an analysis of the relationship between different topics and toxicity.

Despite implicative evidence of the relationship between news topics and online hate, *toxicity of the comments of online news content* has not been systematically analyzed *by news topic* in previous research. It is this research gap that we aim to address. We specifically investigate a concept that we refer to as *online news toxicity*, defined as toxic commenting taking place in relation to online news. Our aim is to analyze if different topics result in varying levels of toxic commenting. For this, we pose the following research questions:

- **RQ1**: How does online news toxicity vary by news topic?

- **RQ2**: What are the key themes characterizing online news toxicity?

To address these questions, machine learning provides value, as it facilitates dealing with large-scale online data [11,23]. We address RQ1 by collecting a large dataset of YouTube news videos and all comments of those videos. We then topically classify the stories using supervised machine learning, and score each comment using a publicly available toxicity scoring service that has been trained using millions of social media comments. Using these two variables–toxicity and topic–we quantitatively analyze how toxicity varies by news topic. To address RQ2, we conduct an in-depth qualitative analysis of the relationship between content type and toxicity. We conclude by discussing the implications for journalists and other stakeholders and outlining future research directions.

The focus on the online news context is important for a variety of reasons. First, because of the impact that news stories have in the society in shaping citizen's worldview and the quality of public discourse [24]. Second, understanding toxic responses to online news stories matters to many stakeholder groups within the media profession, including online news and media

organizations, content producers, journalists and editors, who struggle to make sense of the impact of their stories on the wider stratosphere of social media.

Third, in the era of mischievous strategies for getting public attention, it is becoming increasingly difficult for news media to provide facts without seen as a manipulator or stake-holder in the debate itself. Previous research on online hate, suggest that toxicity is especially prevalent in online news media [11]. In the present time, news channels cannot isolate them-selves from the audience reactions, but analyzing these reactions is important to understand the various sources of digital bias and to form an analytical relationship to the audience. Finally, the betterment of online experiences by mitigating online toxicity is a matter of socie-tal impact, as toxic conversations impact nearly all online users across social media platforms [10,12,25].

## Literature review

### Antecedents for online toxicity

In online environments, toxic behavior is often seen enhanced by the fact that participants can typically comment anonymously and are not held accountable for their behavior in the same way as in offline interactions [3]. Online communities for marginalized or vulnerable groups are particularly exposed to online toxicity because discriminatory patterns, including sexism and racism, tend to be perpetuated and exacerbated online [26].

While inclusivity, accessibility and low barriers to entry have increased individual and citi-zen participation and the associated public debate on matters of social importance, toxic dis-cussions show the cost of having low barriers or supervision for online participation. Because everyone can participate, also the people with toxic views are participating. Some studies high-light democracy of online environments as a contributing factor of online controversies [4,27]. Because the Internet brings together people with different backgrounds and allows a space for people to interact that do not normally interact with each other, an environment is created where contrasting attitudes and points of view are conflicting and colliding.

Another explanation for online toxicity is that, even though online environments give unprecedented access to differing views and information, people tend to actively filter out information that is contrasting their existing views [21] and seek the company of like-minded individuals, forming closed "echo chambers". These echo chambers are environments where like-minded people reinforce each other's views, either without exposure to the views of the opposing side or seeing these views as the target of ridicule from the perspective of the shared narrative of the community [28]. Furthermore, the echo chambers may result in group polari-zation, in which a previously held moderate belief (e.g., "I'm not sure about the motives of the refugees") is taking a more extreme form following the more radical elements of the commu-nity (e.g., "refugees are not really escaping violence but to get free social benefits").

A fundamental question that scholars investigating online hate are asking is whether online environments lend themselves *sui generis* to provocative and harassing behavior. Khorasani [29] notes that, like their counterparts in actual social networks, participants in online groups "make friendships and argue with each other and become involved in long and tedious con-flicts and controversies" (p. 2). Moule et al. [30] observe, however, that online environments have created new forms of socialization and have forged changes in intra- and inter-group relations. Hardaker [3] argues that the relative anonymity provided in online exchanges "may encourage a sense of impunity and freedom from being held accountable for inappropriate online behaviour" [sic] (p. 215). In a similar vein, Chatzakou et al. [31] observe that because of the pseudo-anonymity of online platforms, people tend to express their viewpoints with less inhibition than they would in face-to-face interactions. Patton et al. [5] note the reciprocal

relationship between online and offline violence. The low barriers of entry of online environments, they argue, have changed how peer-to-peer relationships are managed [5]. In sum, these previous findings support and stress the need for research on online toxicity.

## Topics and online toxicity

Prior research has found that certain topics are more controversial than others (see Table 1). These include nationalism [29,32], sexism [31], agricultural policies [33], climate change (*ibid.*), religious differences (*ibid.*), defense [34], foreign policy (*ibid.*), intelligence agencies (*ibid.*), politician's characteristics/personality traits (*ibid.*), energy [35], vaccination [19], fake news [19], and gun control [26,34]. For example, Kittur et al. [27] found that Wikipedia articles on well-known people, religion and philosophy involved more controversy and conflict.

In general, the intersects between users' commenting behaviour and the topic of news items are not yet well understood, even though some studies on negative user behavior explicate the link between topics and toxic commenting. It has been found that although controversial political or social topics typically generate more user comments, users often read news comments for their entertainment value rather than in response to the news article itself [43]. Another study found that writers of toxic comments rearticulated the meaning of news items to produce hate against a marginalized group, even if that group was not the topic of the news [44].

Although existing research on negative online behavior has implications for the research questions posed in this study, the relationship between online news topics and the toxicity of user comments has not been studied directly and systematically. The closest study we could locate is by Ksiazek [34] who offers a content analysis of news stories and user comments across twenty news websites with the aim of predicting the volume of comments and their relative quality in terms of civility and hostility. Hostility was defined as comments "intentionally

**Table 1. Topics for online toxicity.**

| Topic for toxicity | Definition / examples | Reference |
|---|---|---|
| Consumer firestorms | Consumer criticism toward corporations (e.g., Facebook outcry about a company's billboard ads; Facebook privacy issues; Korean airlines firestorm; NFL's CoverGirl ad; Notebook brand Moleskin asked designers to submit "free" designs; NYPD and McDonalds asking consumers to make positive online posts) | [36] [33] [4] |
| Environment | Polarizing environmental issues (e.g., climate change, agricultural policies, wind energy, biofuels, the Fukushima disaster) | [35] [33] [19] |
| Health | Health related commenting (e.g., vaccine controversies, food security) | [19] |
| Interpersonal | Disagreements between active members of specialized online discussion forums (e.g., petty disputes in a community forum) | [29] [3] |
| Media | Media and online platforms (fake news; fake reviews of tourist destinations and hospitality businesses) | [37] [19] |
| People | Personal attacks against public figures and well-known people (e.g., Woody Allen, Trump, attacking memorial pages of deceased people, known as RIP trolling) | [38] [39] [40] [11] |
| Philosophy | Philosophical debates | [40] |
| Politics | Political issues (Wikileaks and Edward Snowden, gun rights/gun control, news stories relating to economy, government inefficiency, immigration, defense, foreign policy, intelligence agencies, and politicians' personality traits) | [33] [19] [26] [34] |
| Race | Race-related commenting (e.g., racist abuse on Twitter of an FA football player) | [41] [38] |
| Religion | Religious differences (e.g., Islamophobia) | [33] [40] |
| Sexism | Gender-related commenting (e.g., the #gamergate controversy related to gaming culture) | [42] [33] [26] |

designed to attack someone or something and, in doing so, incite anger or exasperation through the use of name-calling, character assassination, offensive language, profanity, and/or insulting language" [34] (p. 854). The study found that news stories about the economy, government inefficiency, immigration, gun control, defense, foreign policy, intelligence agencies, and politicians' personality traits are more prone to elicit hostile discussion.

Several other studies have treated the relationship between topic and toxicity implicitly. Wang and Liu [45] find support for readers' varied emotional reactions specifically to news articles, while Salminen et al. [11] analyze the targets of online hate and find that media is targeted frequently in their dataset. Drawing on sociolinguistics and the social pragmatics of politeness, Zhang et al. [22] study some of the "warning signs" in the context of English Wikipedia that may indicate that an online conversation that started civil is about to derail. However, their study is explicitly topic-agnostic, as it disregards the influence of topic and focuses solely on the presence of rhetorical devices in online comments.

Most notably, these earlier studies did not perform a topical analysis of the content. To extend the online research toxicity, we conduct a topical analysis to better understand the audience's toxic responses to online news content. Although the relationship between news topics and online toxicity has not been systematically investigated, the broader literature on online hate speech suggests that topic sits within a host of other factors, all of which contribute to understanding the phenomenon of toxicity in online commenting. These studies point to the need for a deeper analysis of the intersects of personal values, group membership, and topic. While this study focuses only on the relationship between topic and toxicity, it is conducted with the understanding that the results provide a springboard for further research on the complex nature of toxic online commenting.

## Methodology

### Research design

We use machine learning to classify the topics of the news videos. For this, we use a fully connected Feed-Forward Neural Network (FFNN) that is a simple and widely used classification architecture [46]. We then score the toxicity of the comments automatically using a publicly available API service. The use of computational techniques is important because the sheer number of videos and comments makes their manual processing unfeasible. In this research, we utilize the website content, tagged for topics, to automatically classify the YouTube videos of the same organization that lack the topic labels. In other words, the FFNN is trained on textual articles from organization's website, which are tagged with topic labels, and then used to predict the topics of YouTube videos, using their titles and descriptions. To answer our research question, we need to classify the videos because videos include user comments whose toxicity we are interested in. We then score each comment in each video for toxicity and carry out statistical testing to explore the differences of toxicity between topics. Additionally, we conduct a qualitative analysis to better understand the reasons for toxicity in the comments.

### Research context

Our research context is Al Jazeera Media Network (AJ), a large international news and media organization that reports news topics on the website and on various social media platforms. Overall, AJ is a reputable news organization, internationally recognized for its journalism. However, from the content, we can see that the channel's content has a "liberal" undertone that can be associated with political polarization between right and left, especially prominent in social media in the wake of the US presidential campaign in 2016 [47]. Previous research on toxicity in the organization's social media comments [11] has shown that AJ's content attracts

a high number of toxic commenting. This can partly be explained by the fact that the audience consists of viewers from more than 150 countries, forming a diverse mix of ethnicities, cultures, social and demographic backgrounds. Previous literature implies that such a mix likely results in conflicts. At the same time, the organization represents an interesting research context as it reports news on a wide range of serious topics and is not geographically restricted–for example, AJ covers US politics but also international politics, European affairs and so on. However, this excludes entertainment and sports (apart from major sports events such as World Cup of football). For this reason, we characterize the content as "serious news" and consider the wide range of topics and diversity of the audience as well as the associated high prevalence of toxic commenting suitable for the purpose of this study.

## Data collection

We collect two types of data from the news content (see Table 2): text content from news stories published in English on Al Jazeera's (a) website (https://www.dropbox.com/s/keccjwuz0ruyztt/website%20data%20collection%20script.txt?dl=0) and (b) AJ+, one of the organization's YouTube channel (https://www.youtube.com/channel/UCV3Nm3T-XAgVhKH9jT0ViRg). The website has more than 15M monthly visits, and the YouTube channel has more than 500,000 subscribers (August 2019).

For YouTube data collection, we use the official YouTube Analytics API (https://developers.google.com/youtube/analytics/) with the channel owner's permission and in compliance with YouTube's terms of service. From YouTube, we retrieve all 33,996 available (through September 2018) videos with their titles, descriptions, and comments. The comments in this channel are not actively moderated, which provides a good dataset of the unfiltered reactions of the commentators. We collect the news stories using a Python script that retrieves the HTML content of new stories from the news organization's website (see S1 File), including information about the article's content, title, publication date, and topics. The website data contains 21,709 news articles, of which 13,058 (60.2%) have been manually tagged by AJ's journalists and editors for topical keywords. Overall, there are 801 topical keywords used by the journalists to categorize the news articles. This tagging is done to improve the search-engine indexing of the news stories, so that the tags are placed in the content management system upon publishing the news story to characterize the content with topically descriptive tags, such as "racism", "environment", "US elections", and so on.

## Data pre-processing

The HTML content from the website contains some unnecessary information for the classification task, such as JavaScript functions, file directories, hypertext marking (HTML), white spaces, non-alphabetical characters, and stop words (i.e., common English words such as 'and', 'of' that provide little discriminative value). These add no information for the classifier algorithm and are thus removed. As machine learning models take numbers as input [48], we convert our articles into numbers using the *Term Frequency–Inverted Document Frequency* (TF-IDF) technique that counts the number of instances each unique word appears in each content piece. TF-IDF scores each word based on how common the word is in a given content

**Table 2. Description and purpose of data.**

|  | Description | Content | Purpose |
|---|---|---|---|
| *Dataset 1*: *YouTube* | Comments and Video title and description | 33,996 videos | To analyze the toxicity of videos by topic |
| *Dataset 2*: *Website* | News articles (HTML body text, titles), news keywords (topics) | 21,709 webpages | To train the topic classifier for YouTube content |

piece, and how uncommon it is across all content pieces [49]. We then convert the cleaned articles into a TF-IDF matrix, excluding the most common and rarest words. Finally, we assign training data and ground-truth labels using a topic-count matrix.

## News topic classification

We use the cleaned website text content, along with the topics, to train a neural network classifier that classifies the collected videos for news topics. Note that the contribution of this paper is not to present a novel method but rather to apply well-established machine learning methods to our research problem. To this end, we develop an FFNN model using the *Keras*, a publicly available Python Deep Learning library (http://keras.io) that enables us to create the FFNN architecture (a fully connected two-layer network). Additionally, we create a custom class to cross-validate and evaluate the FFNN, since Keras does not provide support for cross-validation by default. This is needed because cross-validation is an important step for ensuring that machine learning results are correct [50].

Training of the FFNN was done using the website data because the journalists have actively labeled the news articles for topics using their content management system that generates the topics as "news keywords" that can be automatically retrieved from the HTML source code. The YouTube content is not tagged, only containing generic classes chosen when uploading the videos on YouTube. The topics created by the journalists are crucial because journalists are considered as subject-matter experts of news, and the use of expert-labeled data generally improves the performance of supervised machine learning [51], because human expertise is helpful for the model to detect patterns from the data. From a technical point of view, this is a multilabel classification problem, as one news article is typically labeled for several topics. Note, however, that for statistical testing we only utilize the highest-ranking topic per a news story. More specifically, the output of the FFNN classifier is a matrix of confidence values for the combination of each news story and each topic. Of these, the chosen topics are the ones exceeding a set threshold value for the confidence–in our case, we use the commonly applied value of 0.5 for testing and, for statistical, we choose the topic with the highest confidence value. In other words, a story has only one "dominant" topic in the statistical analysis. This is done for parsimony, as using all or several topics per story would make the statistical comparison exceedingly complex.

## Classifier evaluation

Here, we report the key evaluation methods and results of the topic classification. Note that a full evaluation study of the applied FFNN classifier is presented in Salminen et al. [48].

First, to optimize the parameters of the FFNN model, we create a helper class to conduct random optimization on both the TF-IDF matrix creation and the FFNN parameters. Subsequently, we identify a combination of FFNN parameters in the search space that provides the highest F1 Score (i.e., the harmonic mean of precision and recall). This combination is used to fine-tune the model parameters, and we obtain a solid performance ($F1_{FFNN} = 0.700$). By "solid," we mean that the results are satisfactory for this study, so that the accuracy of our algorithm is considerably higher than the probability of choosing the right topic by random chance ($p = 1 / 799 \approx 0.1\%$). FFNN also clearly outperforms a Random Forest (RF) model that was tested as a baseline model ($F1_{RF} = 0.458$).

As an alternative to the supervised methods, we also experimented with *Latent Dirichlet Allocation* (LDA), an unsupervised topic modeling approach [52]. LDA infers latent patterns ("topics") from the distribution of words in a corpus [53]. For brevity, we exclude the results of these experiments from the manuscript; a manual inspection showed that the automatically

inferred LDA topics are less meaningful and interpretable as the news keywords handpicked by the journalists working for the organization whose content we are analyzing. Therefore, we do not use LDA but rather train a supervised classifier based on manually annotated data by journalists that can be considered as experts of news topics. The importance of using domain experts for data annotation is widely acknowledged in machine learning literature [54,55]. Generally, expert taxonomies are considered as gold standards for classification [56].

We apply the model trained on website content (i.e., the cleaned article text) is applied to video content (i.e., the concatenated title and description text). Intuitively, we presume this approach works because the news topics covered in the YouTube channel are highly similar to those published on the website (e.g., covering a lot of political and international topics). Because we lack ground truth (there are no labels in the videos), we evaluate the validity of the machine-classified results by using three human coders to classify a sample of 500 videos using the same taxonomy that the machine applied. We then measure the simple agreement between the chosen topics by machine and human raters and find that the average agreement between the three human raters and the machine is 70.4%. Considering the high number of classes, we are satisfied with this result. In terms of success rate, the model provided a label for 96.1% of the content (i.e., 32,678 out of 33,996 YouTube videos).

## Toxicity scoring

Alphabet, Google's parent company, has launched an initiative, the Perspective API, aimed at preventing online harassment and providing safer environments for user discussions via the detection of toxic comments. Perspective API has been trained on millions of online comments using semi-supervised methods to capture the toxicity of online comments in various contexts [1]. Perspective API (https://perspectiveapi.com) defines a toxic comment as "a rude, disrespectful, or unreasonable comment that is likely to make you leave a discussion" [57]. This definition is relevant to our research, since it specifically focuses on online comments of which our dataset consists. Note that Perspective API is a publicly available service for toxicity prediction of social media comments, enabling replicability of the scoring process.

We utilize the Perspective API to score the comments collected for this study. After obtaining an access key to the API, we test its performance. The version of the API at the time of the study had two main types of models: (a) alpha models and (b) experimental models. The alpha models include the default toxicity scoring model, while the experimental models include the severe toxicity, fast toxicity, attack on author, attack on commenter, incoherent (i.e., difficult to comprehend), inflammatory (provocative), likely to reject (according to New York Times moderation guidelines), obscene, spam, and unsubstantial (i.e., short comments). In this research, we use the alpha category's default toxicity model that returns a score between 0 and 1, where 1 is the maximum toxicity. According to the Perspective API's documentation, the returned scores represent *toxicity probability*, i.e., how likely a comment is perceived to be toxic by online users. To retrieve the toxicity scores, we sent the 320,246 comments to Perspective API; however, the tool returned some blank values. According to the API documentation, failure to provide scores can be due to non-English content, and too long comments. Overall, we were able to successfully score 240,554 comments, representing 78.2% of the comments in the dataset.

A manual inspection showed that Perspective API was able to detect the toxicity of the comments well. To further establish the validity of the automatic scoring of Perspective API, we conducted a manual rating on a random sample of 150 comments. A trained research assistant determined if a comment is hateful or not (yes/no), and we compared this rating to the score of Perspective API. We use the threshold of 0.5 so that comments below that threshold are

considered non-toxic and comments above toxic (note that this is comparable to the decision threshold of the classifier, also 0.5). We obtained a percentage agreement of 76.7% between the human annotator and the score given by Perspective API, which we deem reasonable for this study. We also computed Cohen's Kappa that considers the probability of agreeing by chance. In total, there were 135 agreements (90% of the observations), whereas the number of agreements expected by chance would have been 118.5 (79% of the observations). The obtained Kappa metric of $\kappa = 0.524$ indicates a "moderate" agreement [58]. While the score would ideally be higher, we consider it acceptable for this study, especially given the evidence that toxicity ratings are highly subjective in the real world [59,60].

## Obtaining toxicity scores of news topics

After scoring the video comments, we associate each comment with a topic from its video. As the toxicity score of each comment is known, we simply calculate the average toxicity score of the comments of a given video. After this, we have obtained the average toxicity score for each video based on its comments' toxicity. Because we also have the topic of each video classified using the FFNN, taking the average score of all the videos within a given topic returns the average toxicity score of that topic.

## Quantitative analysis

### Data preparation

To simplify the statistical analysis, we reduce the number of classes by grouping similar topics under one theme ("superclass"). Thus, we group people into countries, countries into continents, and similar themes under one topic. In most cases, we kept the original names given by the journalists to the topics, only adding another topic. For example, Environment, Climate SOS and Weather became Environment & Weather. We grouped country names under continents. Many observations for Middle Eastern countries caused the creation of a separate superclass Middle East. Likewise, Israel, Palestine, and Gaza were grouped into the superclass Israel-Palestine. The superclass grouping was done manually by one of the researchers grouping the topics into thematically consistent classes, with another researcher corroborating that the superclasses logically correspond to the original classes. Table 3 shows the superclasses along with the number of topics and news videos in them. S1 Table provides a detailed taxonomy of the grouping.

By creating the superclasses, we reduced 73 topics to 19 superclasses, with a decrease of 74% in terms of the number of classes to analyze. This increases the power of the analysis by increasing the number of observations per class and makes the results easier to interpret.

### Results

Exploring the means of toxicity by superclass reveals interesting information (see Table 4). For example, Racism has the highest average toxicity (M = 0.484, SE = 0.018) out of the news topics, while Science & Technology has the lowest (M = 0.277, SE = 0.007). Out of countries, news stories about Russia have the most toxic responses (M = 0.426, SE = 0.013), while stories about Latin America have the least toxicity (M = 0.359, SE = 0.006).

While explorative results are interesting, we cannot argue that the toxicity of Racism is higher than that of other superclasses without testing if the difference is statistically significant. This testing is done by comparing the average comment toxicity between the superclasses using regression analysis with dummy variables, as shown in Eq 1:

$$CT_i = \beta_0 + \beta_1 * Superclass_i^1 + \cdots + \beta_{19} * Superclass_i^{19} + \varepsilon_i, \tag{1}$$

**Table 3. Superclasses (SC) and sample parameters.** Note that "Israel-Palestine" is considered as a news topic rather than region because the news stories in this category deal with various aspects of the regional conflict.

| | Superclass | Sub-classes in SC | Videos in SC |
|---|---|---|---|
| | *News Topics* | | |
| 1 | Arts & Culture | 1 | 414 |
| 2 | Business & Economy | 1 | 831 |
| 3 | Environment & Weather | 3 | 309 |
| 4 | Health | 1 | 142 |
| 5 | Human rights | 1 | 287 |
| 6 | Israel-Palestine Conflict | 5 | 1012 |
| 7 | Media | 2 | 3054 |
| 8 | Politics | 9 | 1474 |
| 9 | Racism | 1 | 61 |
| 10 | Science & Technology | 1 | 356 |
| 11 | Sport | 2 | 63 |
| 12 | War & Conflict | 6 | 741 |
| | *Countries & Regions* | | |
| 13 | Africa | 4 | 4819 |
| 14 | Asia | 12 | 3338 |
| 15 | Europe | 5 | 4348 |
| 16 | Latin America | 2 | 695 |
| 17 | Middle East | 12 | 5165 |
| 18 | Russia | 1 | 153 |
| 19 | US & Canada | 4 | 2258 |
| | Total | 75 | 29,520 |

**Table 4. Toxicity of superclasses.** Mean indicates average comment toxicity of the videos in the superclass.

| Superclass | Mean toxicity | Std. Err. | 95% CI | |
|---|---|---|---|---|
| *News topics* | | | | |
| Racism | 0.484 | 0.018 | 0.448 | 0.521 |
| Israel-Palestine Conflict | 0.474 | 0.004 | 0.466 | 0.482 |
| War & Conflict | 0.423 | 0.005 | 0.412 | 0.434 |
| Human Rights | 0.395 | 0.009 | 0.377 | 0.412 |
| Media | 0.374 | 0.002 | 0.368 | 0.380 |
| Politics | 0.370 | 0.004 | 0.362 | 0.379 |
| Business & Economy | 0.328 | 0.005 | 0.317 | 0.339 |
| Sport | 0.313 | 0.027 | 0.259 | 0.367 |
| Health | 0.310 | 0.014 | 0.281 | 0.339 |
| Arts & Culture | 0.303 | 0.008 | 0.286 | 0.320 |
| Environment & Weather | 0.301 | 0.009 | 0.283 | 0.320 |
| Science & Technology | 0.277 | 0.007 | 0.261 | 0.292 |
| *Countries & regions* | | | | |
| Russia | 0.426 | 0.013 | 0.400 | 0.451 |
| Middle east | 0.416 | 0.002 | 0.412 | 0.421 |
| Europe | 0.379 | 0.002 | 0.374 | 0.383 |
| US & Canada | 0.376 | 0.003 | 0.370 | 0.382 |
| Asia | 0.371 | 0.002 | 0.365 | 0.376 |
| Africa | 0.370 | 0.002 | 0.365 | 0.375 |
| Latin America | 0.359 | 0.006 | 0.345 | 0.372 |

where $CT_i$ is the average comment toxicity of a news story $i$, belonging to superclass $j$ ($j$ = 1 to 19). Beta is the estimated regression coefficient. Moreover, $Superclass_i^j$ is a dummy variable for superclass $j$. For each pairwise comparison, we exclude one of the dummy variables, which makes it a base category against which all other categories are compared.

Since our regression has no other variables, the coefficient on every dummy variable represents the difference in mean values of toxicity of the respective superclass and the base superclass. Note that the F-test on this regression is equivalent to one-way analysis of variance (ANOVA) test for all groups, with the following hypotheses:

- Null hypothesis: All βj (j > 0) are equal to zero.

- Alternative hypothesis: at least one of the βj (j > 0) differs from zero.

Rejecting the null hypothesis indicates that at least one of the two means are not equal and substantiate further pairwise comparison between means to clarify the exact pattern of differences. Given the regression specification, pairwise comparison of the superclass means–i.e., testing for statistical significance between means of two superclasses–can be done by t-test for statistical significance of respective dummy coefficients.

However, the consistency and efficiency of the coefficients' estimation by the ordinary least squares (OLS) method is based on the viability of several assumptions. One of the most vulnerable assumptions is equality of variance of the error term $\varepsilon_i$ across the observations. The Cook-Weisberg [61] test for heteroskedasticity shows its violation for our dataset. Hence, we apply the Huber–White estimator of variance, which is a heteroskedasticity-robust estimation procedure [62]. Another aspect of validity in pairwise comparisons is the adjustment of p-values to account for multiple comparisons. These adjustments are needed because we perform the tests simultaneously on a single set of data. As a matter of sensitivity analysis, three types of adjustments are applied here: Bonferroni, Sidak, and Scheffe [63]. S2 Table shows the pairwise comparisons with each of these adjustments. Fig 1 shows a summary of conclusive (red in

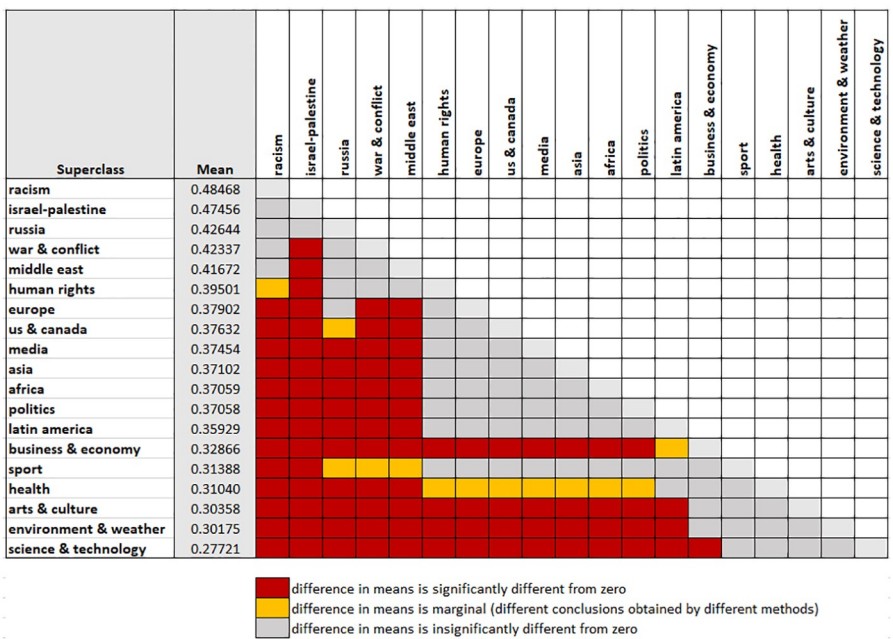

**Fig 1. Toxicity differences between topics.** Red indicates differences that are robust across the applied three multiple comparison tests. Orange indicates differences where the multiple comparison tests give inconclusive results, and grey cells are differences that not significant at p = 0.05.

Fig 1) and inconclusive (yellow in figure) results. Due to the high number of pairwise comparisons, we show the results in the form of a matrix, where color indicates the significance of the mean differences.

From the results, we observe that four topics have consistently fewer toxic responses: Science & Technology, Environment & Weather, Arts & Culture, and Business & Economy. Two other categories, Sport and Health, are also less provocative, although some of the test methods return insignificant results. The main reason for the inconclusive results for these two categories is likely their smaller number of observations. More provocative topics, in comparison to others, are Israel-Palestine, War & Conflict, Middle East, Russia, and Racism. The data along with full statistical results is available in S2 File.

## Qualitative analysis

In addition to the quantitative analysis, we perform a qualitative analysis on a smaller subset of videos and the comments belonging to those videos. For this analysis, we purposefully sample 30 videos with the highest and 30 videos with the lowest toxicity scores from 9 video categories (5 of them being *more toxic*: Racism, Israel-Palestine, Russia, War & Conflict, and Middle East; and 4 of them being *less toxic*: Science & Technology, Environment & Weather, Arts & Culture, and Business & Economy). This sampling results in $30 \times 2 \times 9 = 540$ videos.

These 540 videos and their comments were analyzed for analytical questions (AQs):

- AQ1: Why are the comments likely to be toxic in a given superclass?

- AQ2: When are the comments in a generally toxic topic non-toxic? (For AQ2, we wanted to carry out a comparison between toxic and non-toxic videos–while a topic can raise a lot of toxicity, can we find cases where the comments are considerably less toxic? If so, what is the reason for that?)

- AQ3: When are the comments "toxicified"? (That is, when and why does a neutral topic like sport become toxic?)

To address these questions, one of the researchers browsed all the sampled videos manually, examining their content, and reading the associated comments on YouTube. This researcher identified themes to address the analytical questions. Another researcher investigated the theme taxonomy and corroborated it. After this, the first researcher completed the analysis.

We also recorded the number of views, duration, number of likes and dislikes, and the number of comments for each analyzed video. This manual data collection was performed during the final two weeks of April 2019, with the statistics for this data given in Table 5.

To understand whether the number of views, duration, number of likes and dislikes, and the number of comments are indicative of the toxicity score of a video, we calculated the Pearson coefficient between these values. The significant results are shown in Table 6. Although there does not seem to be a strong unifying story, it appears that *more dislikes to a video and a greater number of comments correlate with more toxic video discussions*, while *more likes*, *a greater number of views*, *and longer videos correlate with less toxicity*. While the correlation for likes vs. dislikes and the number of views with video toxicity score are easy to explain, duration is a surprising factor. Seemingly, *the longer the video, the less toxic its discussions are likely to be*. This leads us to believe that, perhaps, users did not want to comment without watching the entirety of a video and when the videos were longer, this probably dissuaded them from watching the content and commenting.

**Table 5. Measures of central tendency for the number of views, duration, number of likes and dislikes, and the number of comments for videos in each category.**
The table ignores the missing values of the videos that were removed between the collection of quantitative and qualitative data.

| Category | # of Views | Duration (secs) | # of Likes | # of Dislikes | # of Comments |
|---|---|---|---|---|---|
| Racism Low | $\bar{x} = 9628.47$<br>$s = 16056.94$<br>$R = 364;71998$ | $\bar{x} = 575.47$<br>$s = 896.01$<br>$R = 116;2871$ | $\bar{x} = 106.93$<br>$s = 203.97$<br>$R = 2;982$ | $\bar{x} = 21.53$<br>$s = 34.42$<br>$R = 0;162$ | $\bar{x} = 33.73$<br>$s = 38.52$<br>$R = 1;153$ |
| Racism High<br>(1 missing value) | $\bar{x} = 6426.24$<br>$s = 8128.35$<br>$R = 683;31889$ | $\bar{x} = 356.07$<br>$s = 666.66$<br>$R = 51;2998$ | $\bar{x} = 61.76$<br>$s = 75.37$<br>$R = 5;318$ | $\bar{x} = 17.00$<br>$s = 23.67$<br>$R = 0;89$ | $\bar{x} = 59.72$<br>$s = 96.30$<br>$R = 0;353$ |
| Israel-Palestine Low<br>(6 missing values) | $\bar{x} = 2633.30$<br>$s = 3701.22$<br>$R = 87;16839$ | $\bar{x} = 338.71$<br>$s = 665.48$<br>$R = 6;2852$ | $\bar{x} = 25.71$<br>$s = 21.45$<br>$R = 1;80$ | $\bar{x} = 3.67$<br>$s = 4.34$<br>$R = 0;16$ | $\bar{x} = 6.71$<br>$s = 8.36$<br>$R = 1;37$ |
| Israel-Palestine High<br>(2 missing values) | $\bar{x} = 1597.36$<br>$s = 2884.00$<br>$R = 86;15594$ | $\bar{x} = 189.21$<br>$s = 269.48$<br>$R = 19;1500$ | $\bar{x} = 13.43$<br>$s = 12.08$<br>$R = 1;62$ | $\bar{x} = 3.86$<br>$s = 7.13$<br>$R = 0;37$ | $\bar{x} = 5.71$<br>$s = 7.26$<br>$R = 0;32$ |
| Russia Low | $\bar{x} = 4021.80$<br>$s = 4424.29$<br>$R = 132;14901$ | $\bar{x} = 237.67$<br>$s = 357.12$<br>$R = 30;1582$ | $\bar{x} = 26.67$<br>$s = 23.60$<br>$R = 0;127$ | $\bar{x} = 7.30$<br>$s = 10.57$<br>$R = 0;42$ | $\bar{x} = 10.33$<br>$s = 12.71$<br>$R = 0;61$ |
| Russia High<br>(1 missing value) | $\bar{x} = 2191.59$<br>$s = 2435.35$<br>$R = 137;13096$ | $\bar{x} = 321.31$<br>$s = 697.42$<br>$R = 26;3673$ | $\bar{x} = 15.86$<br>$s = 11.09$<br>$R = 3;58$ | $\bar{x} = 6.45$<br>$s = 8.33$<br>$R = 0;44$ | $\bar{x} = 9.72$<br>$s = 11.38$<br>$R = 0;39$ |
| War & Conflict Low<br>(1 missing value) | $\bar{x} = 2734.83$<br>$s = 2610.78$<br>$R = 542;12276$ | $\bar{x} = 271.48$<br>$s = 377.47$<br>$R = 57;1500$ | $\bar{x} = 19.24$<br>$s = 16.25$<br>$R = 4;69$ | $\bar{x} = 5.62$<br>$s = 6.29$<br>$R = 0;21$ | $\bar{x} = 3.34$<br>$s = 3.73$<br>$R = 1;16$ |
| War & Conflict High<br>(2 missing values) | $\bar{x} = 2638.11$<br>$s = 1963.07$<br>$R = 606;8245$ | $\bar{x} = 137.36$<br>$s = 109.22$<br>$R = 39;658$ | $\bar{x} = 18.32$<br>$s = 28.19$<br>$R = 3;159$ | $\bar{x} = 3.89$<br>$s = 3.64$<br>$R = 0;17$ | $\bar{x} = 4.46$<br>$s = 5.32$<br>$R = 0;25$ |
| Middle East Low<br>(6 missing values) | $\bar{x} = 2278.25$<br>$s = 2057.82$<br>$R = 408;10329$ | $\bar{x} = 380.67$<br>$s = 528.34$<br>$R = 30;1616$ | $\bar{x} = 14.29$<br>$s = 13.55$<br>$R = 3;64$ | $\bar{x} = 2.88$<br>$s = 3.37$<br>$R = 0;14$ | $\bar{x} = 2.54$<br>$s = 3.19$<br>$R = 0;16$ |
| Middle East High<br>(3 missing values) | $\bar{x} = 4630.85$<br>$s = 15611.16$<br>$R = 392;82609$ | $\bar{x} = 202.67$<br>$s = 278.34$<br>$R = 15;1500$ | $\bar{x} = 9.63$<br>$s = 5.36$<br>$R = 4;30$ | $\bar{x} = 2.82$<br>$s = 3.41$<br>$R = 0;13$ | $\bar{x} = 1.04$<br>$s = 0.81$<br>$R = 0;3$ |
| Science & Technology Low (3 missing values) | $\bar{x} = 2585.33$<br>$s = 2024.39$<br>$R = 546;9105$ | $\bar{x} = 896.33$<br>$s = 942.85$<br>$R = 76;2888$ | $\bar{x} = 23.48$<br>$s = 14.73$<br>$R = 2;52$ | $\bar{x} = 2.67$<br>$s = 4.69$<br>$R = 0;24$ | $\bar{x} = 2.37$<br>$s = 1.88$<br>$R = 0;9$ |
| Science & Technology High (3 missing values) | $\bar{x} = 6261.59$<br>$s = 15223.10$<br>$R = 340;73862$ | $\bar{x} = 1301.74$<br>$s = 1002.70$<br>$R = 19;2362$ | $\bar{x} = 48.26$<br>$s = 122.39$<br>$R = 2;642$ | $\bar{x} = 12.74$<br>$s = 27.25$<br>$R = 0;107$ | $\bar{x} = 37.48$<br>$s = 109.88$<br>$R = 0;563$ |
| Environment & Weather Low (2 missing values) | $\bar{x} = 5316.36$<br>$s = 6722.71$<br>$R = 277;25445$ | $\bar{x} = 505.54$<br>$s = 533.89$<br>$R = 72;1500$ | $\bar{x} = 42.00$<br>$s = 50.45$<br>$R = 2;220$ | $\bar{x} = 2.86$<br>$s = 3.62$<br>$R = 0;16$ | $\bar{x} = 4.50$<br>$s = 4.26$<br>$R = 0;14$ |
| Environment & Weather High | $\bar{x} = 1999.97$<br>$s = 3101.94$<br>$R = 471;14705$ | $\bar{x} = 188.03$<br>$s = 249.00$<br>$R = 85;1500$ | $\bar{x} = 13.47$<br>$s = 7.49$<br>$R = 2;34$ | $\bar{x} = 2.43$<br>$s = 2.60$<br>$R = 0;10$ | $\bar{x} = 6.37$<br>$s = 7.51$<br>$R = 1;34$ |
| Arts & Culture Low (1 missing value) | $\bar{x} = 1963.66$<br>$s = 1710.40$<br>$R = 276;6709$ | $\bar{x} = 289.62$<br>$s = 418.68$<br>$R = 99;1511$ | $\bar{x} = 22.41$<br>$s = 20.95$<br>$R = 4;80$ | $\bar{x} = 1.07$<br>$s = 1.33$<br>$R = 0;4$ | $\bar{x} = 1.93$<br>$s = 1.65$<br>$R = 1;7$ |
| Arts & Culture High | $\bar{x} = 4381.90$<br>$s = 13052.42$<br>$R = 428;73006$ | $\bar{x} = 132.63$<br>$s = 26.02$<br>$R = 53;188$ | $\bar{x} = 21.43$<br>$s = 19.39$<br>$R = 1;89$ | $\bar{x} = 4.73$<br>$s = 7.21$<br>$R = 0;39$ | $\bar{x} = 5.10$<br>$s = 10.57$<br>$R = 0;56$ |
| Business & Economy Low (1 missing value) | $\bar{x} = 2372.83$<br>$s = 2202.32$<br>$R = 515;12374$ | $\bar{x} = 323.86$<br>$s = 484.23$<br>$R = 21;1560$ | $\bar{x} = 17.90$<br>$s = 11.42$<br>$R = 1;52$ | $\bar{x} = 2.72$<br>$s = 3.38$<br>$R = 0;15$ | $\bar{x} = 2.45$<br>$s = 2.18$<br>$R = 1;9$ |
| Business & Economy High | $\bar{x} = 1798.88$<br>$s = 1239.67$<br>$R = 465;5625$ | $\bar{x} = 335.70$<br>$s = 477.47$<br>$R = 108;1560$ | $\bar{x} = 12.93$<br>$s = 8.01$<br>$R = 4;38$ | $\bar{x} = 2.67$<br>$s = 2.77$<br>$R = 0;10$ | $\bar{x} = 5.23$<br>$s = 4.56$<br>$R = 0;17$ |

**Table 6. Pearson correlation tests and direction between the toxicity score of a video and the number of views, duration, number of likes and dislikes, and the number of comments.**

| Category (Toxicity Scores) | # of Views | Duration (secs) | # of Likes | # of Dislikes | # of Comments |
|---|---|---|---|---|---|
| Racism | - | - | - | - | - |
| Israel-Palestine | - | - | $p < 0.01$ (-) | - | - |
| Russia | - | - | - | - | - |
| War & Conflict | - | $p < 0.05$ (-) | - | - | - |
| Middle East | - | - | - | - | $p < 0.05$ (-) |
| Science & Technology | - | - | - | $p < 0.05$ (+) | - |
| Environment & Weather | $p < 0.05$ (-) | $p < 0.01$ (-) | $p < 0.05$ (-) | - | - |
| Arts & Culture | - | - | - | $p < 0.05$ (+) | - |
| Business & Economy | - | - | $p < 0.05$ (-) | - | $p < 0.01$ (+) |

Reading through and coding the comments and discussions under the videos, it was possible to discover several themes on the emergence of toxicity in these videos. These themes are discussed in the following.

### Graphic videos

Qualitatively watching the videos revealed that graphic videos (typically these videos also have titles and thumbnails that indicate possible graphic content) spark more passionate and accordingly more toxic discussions. In contrast, videos that feature interviews and in-studio commentary pieces have less toxic discussions. Some examples of these graphic videos with high toxicity include *Palestinians fight with Israeli security forces* (BgplkpJrQXg), *Clashes follow Palestinian teen's funeral* (E-ypG-hh4qc), and *Russian troops enter Crimea airbase* (EZzwv2byV6c). In contrast, when an interview or an in-studio commentary has toxicity (e.g., *UpFront—Headliner: Richard Barrett*, ihvq4IlTfFk), it is usually directed toward the presenter or the commentator (e.g., "Idiot [. . .] what you suggest").

### Humanistic stories

Humanistic stories, i.e., ones that tell a story of an individual person, are less likely to attract toxicity, even under categories that are generally toxic like Middle East and War & Conflict. Some examples are *Para athletic championship held in Middle East for first time* (y0Nr4gr6vZQ), *Former Uganda child soldiers return home* (oMFk-jNXZEQ), *Bomb-rigged homes delay return of Iraqi residents near Mosul* (vDN5c7LTb94), and *Ugandan families remember lost children* (Se5KKIRsGH0). Even though there are political framings in these stories that elicit toxicity in other context, civil stories of war and conflict seem to attract less toxic comments. This observation is also in line with previous research by Jasperson and El-Kikhia [64] that underlined the importance of the media organization's role in the humanitarian coverage of the Middle East in American media, especially CNN.

### History and historical facts

Another major source of toxicity was the discussions around historical events and facts. This trend was even more apparent coupled with coverage on underrepresented communities that appear less in English news sources. It is possible to surmise that since English content about these issues appear less in news channels, they attract larger attention and discussion from users who have stakes about the content. Previous research asserts that social media users are more likely to access and share news from international news outlets [65]. Then, it seems likely

that users who feel underrepresented in English news content are likely to disseminate these stories in social media, attracting more traffic and discussion. For example, videos titled *Visiting the first free black town of the new world in Colombia* (8gaXfr9WNwo), *Afro-Cubans still at mercy of white wealth* (9ycZwyIFDHI), *Colombia*: *FARC rebels to disarm at transition zones* (CGy9vVJDsmQ), and *Thailand invites crown prince to become new king* (eCm1LY3z7Kw) follow this trend. The discussions under these videos dominantly take place between locals (rather than locals vs. outsiders) while they are trying to agree upon the events and facts that led to the situations covered in the news piece. These are passionate discussions in English rather than in the local language. From the language use, content, and directions of the discussions we observe that they are made to create a "truthful" representation of events and the community to the international viewers.

## Media as a manipulator

A common trend in less toxic categories like Business & Economy and high toxic categories like Racism is to frame international media as a tool of manipulation and propaganda. This is prevalent even when the message seems acceptable by the viewer (e.g., a comment in the video *STUART HALL—Race*, *Gender*, *Class in the Media*, FWP_N_FoW-I, reads: "Good message but shame it pushes an agenda."). Especially, the coverage of #BlackLivesMatter and related news (e.g., Ferguson shooting) meets with a resistance that frames the organization's coverage as anti-US propaganda that aims to destabilize the US public. Accordingly, this creates friction between users (presumably US citizens) who support these causes and those who see it as a manipulation regardless of the message. Similar discussions arise around discussions regarding Russia and Ukraine—from both sides depending on the context of the video. In a video about Mosul (*Battle for Mosul*: *Iraqi forces advance on eastern front*, ivXrlDpjlB8), users even try to deconstruct the content of the video as well as the political manipulation it aims for (e.g., "the guy on 1.30 is not even Iraqi.").

This trend becomes interestingly apparent in Business & Economy category. Although, generally, the category is a less toxic one, most of its coverage includes a resistance from users who have stakes in the content. For example, in the video *Lebanon's economy affected by Syrian conflict* (CIcNhnQigvU), self-reported Lebanese users paint the coverage as economic manipulation. Similar discussions are in videos *Japan braces for rise in sales tax* (BvWvrp7VZL8), *North Dakota Native Americans feel oil price pinch* (VBLgARaM0Dk), *Cuban economy faces hard times amid fears of Venezuela fallout* (O_BI3p6eNIc), and *Crimea vote brings economic uncertainty* (GJpo6BVaRw4).

## Religion

The final source of toxicity to note are the religious discussions that spark in the comments. They can be framed in two ways: (1) discussions between two users who are of different religious beliefs; (2) discussions between users with and without religious beliefs.

An example for the first category would be this abbreviated exchange between two viewers from the video *Philippines army clashes with rebels in the south* (aBmrw5HEu48):

- User 1: "Islam is a crime against humanity [. . .] Reject Islam and you might just get a taste of peace one day [. . .]"

- User 2: "The Christians wiped out 100M natives in the New World, which is a genocide. A crime against Humanity, The Islamic World never reached that toll, and you say this is a crime against Humanity? How foolish [. . .]"

These discussions are generally framed around the perceived crimes committed by religious institutions and the members of particular religions in the past.

The second category is sparked by user comments, which are non-toxic in nature and covers a sentimental religious adjuration. Frequently, these are met with anger from users (who might be less religious, have no religious beliefs, have a different perspective of the particular religion, or from other religions) who point out that the religious institutions and beliefs were the culprits of these problems in the first place. Here is an example exchange for this category from video *Fragile truce broken in Syria refugee camp* (Tk93DoL67c8):

- "Allah bless mujahideens"

- "Allah does not say to you to pick up arms [. . .] Allah does not say to you to have 20 children and then fail to educate them."

## Discussion

### Positioning findings to prior research

Our findings support the previous research highlighting the impact of topics on the emotional level of user comments in social media [33,40]. We extend this connection to the domain of online news media by specifically focusing on the relationship between online news content and toxicity of social media comments. The topics that are associated with a higher degree of toxicity can be interpreted as more divisive for the online audience, which is accentuated in the online environment that consists of participants with very different backgrounds, cultures, religions, and so on. In general, topics with political connotations (e.g., War & Conflict, Middle-East) arouse more toxicity than non-political topics (e.g., Sports, Science & technology), which corroborates previous research linking politics and online toxicity [11,21,66].

Regarding the qualitative analysis, the association between graphic content and toxicity is in line with previous research which asserts that graphic and/or violent images in news coverage spark a higher interest and elicit more passionate reactions—both negative and positive [67,68]. Multimedia news items elicit more user comments, and there is a small positive correlation between multimedia and online hostility [69]. It has also been suggested that especially carefully framed war imagery has the potential to construct narratives within official agendas and discourse [70]. Then, it becomes possible that these videos spark reactions both to their content and to the agendas that they seem to be developing.

Another specific aspect to mention from the qualitative analysis is that the toxic comments often focus on the topic (e.g., religion, politics), rather than other participants or some unrelated targets. This characterizes the typical nature of toxicity in news context as "topic-driven toxicity" as opposed to other forms of toxicity, such as vindictive toxicity [28] where participants attack against one another. These personal attacks are more common when the participants are interacting; e.g., editing a Wikipedia page with controversial content [1], but they do not seem to be highly prevalent in online news toxicity. This suggests that users are not viewing news video commenting as a collaborative effort (e.g., discussion, conversation, or debate) but just as "an event to comment upon". In particular, attacks against marginalized or vulnerable groups (e.g., minorities, women) that are reported in some earlier studies [26] are seldom present in online news toxicity; again, this highlights the target of toxicity being the "topic" rather than random individual or groups. However, we can observe group-related behavior when the topic is related to a specific group; for example, immigration videos do attract anti-immigration comments and religion videos anti-Islamic commenting.

Moreover, the emergence of the "*History and historical facts*" theme shows how different groups are, in a way, "fighting over the narrative," i.e., how the news stories should be framed and interpreted. This is interestingly contrasting the agenda-setting theory in that the audience may attack the news channel itself, challenging its agenda-setting authority. This conclusion is supported by the "media as a manipulator" theme and may be understood by keeping in mind that online readers fall broadly into "soldiers" (whose online activities are organized and group-based) and "players", "watchdogs" and "believers" (who, for various reasons, act on their own initiative) [43]. In addition, there are obvious linkages to the "fake news" theme, where social media users are increasingly questioning the credibility of news channels [71].

Together, these themes suggest the audience is imposing their own interpretation and views over what happened, rather than readily adopting the facts or the story framing of the focal news outlet. This has at least two important implications: one, for public policy, these comments provide excellent material for analysis of alternative facts or narratives, as social media commentators are clearly voicing their–sometimes deviating–interpretations. Second, the news outlet can use these comments to segments the audience based on the different world-views that are shown in the comments. One approach to this is creation of audience personas using social media data [72,73] or other forms of online news analytics [24].

## Practical implications

In the era of social media, it is becoming increasingly difficult for news media not to be seen as a manipulator or stakeholder in the debate itself. However, the news channels cannot isolate themselves from the audience reactions in the wild. Analyzing new audience's sentiment is important to leverage the two-directional nature of online social media [74] and to understand the various sources of "digital bias" of audiences and the news channels themselves.

Our results suggest that news channels both *have* and *have not* power on the toxicity of the comments in their stories. In summary, the power comes from the fact that both topic selection (i.e., what topics are reported) and topic framing (i.e., how the topic is reported) impact toxicity of the social media commentators' response. Both the empirical findings and the theoretical association between toxicity and agenda setting [75] and online toxicity suggest that content creators–intentionally or unintentionally–have power over the toxicity of online conversations. However, the unpredictable nature of social media commenting can reduce the channel owner's power to govern the comment toxicity. For example, a neutral topic can become "toxicified" after introducing controversial elements, such as religion. We observed examples of this under Israel-Palestine and War-Conflict videos, where different political or national allegiances trigger toxicity, much similar to group polarization behavior [21].

For news channels, to avoid sensitive topics due to likely toxic reactions would be to submit to "tyranny of the audience," i.e., avoiding important topics out of fear for toxic reactions. Obviously, this is not a good strategy, as responsible editorial decisions should be made based on the relevance of news rather than their controversial nature. However, being ignorant of the news audience's reactions is not helpful either, as social media comments nowadays represent a major form of public discourse that the media should not ignore. Therefore, one needs to strike a balance towards fostering a constructive discussion and debate over topics, without sacrificing the coverage of sensitive topics.

Perhaps a useful guideline is that, in the process of topic selection, *content creators should be aware of the content topic's inflammatory nature and possibly use that information to report in ways that mitigate negative responses rather than encourage them*. This approach is compatible with the idea of "depoliticizing" suggested by Hamilton [76]. Note that depoliticizing does not mean avoiding political topics. It means defusing a controversial topic by using a framing

style that is aimed at defusing toxicity while maintaining. In practice, journalists could use information from previous toxicity on a given topic when framing their news stories, especially in the context of topics with known high toxicity.

Especially when dealing with an international audience base, the diversity of religious and political views is likely to result in heightened toxicity when stories are reported in a way that seems unfair or unbalanced for a group of participants. Therefore, we suggest that *content creators should strive for a reporting style that appears objective and balanced, especially for the topics with a history of higher toxic commenting*. To illustrate, consider a binary choice: given the journalist knows Topic A is controversial, does their story framing strategy aim to (a) exacerbate controversy or (b) alleviate controversy? This strategic choice, we argue, is important for the toxicity outcome.

Our qualitative results suggest that when a story belonging to a topic with high average toxicity receives non-toxic responses, this is often consequence on how it is reported. This is especially visible in videos with tags "Humanistic" or "Humanistic stories" that report stories focused on real everyday people. A user quote on the story "Ugandan families remember lost children" sheds light to why humanistic stories are likely to be received more positively: "This is a really great video–informative and easy to watch makes you ponder on how grateful you really are." Overall, toxicity seems less prevalent in these human stories. Note that we do not make the argument that human story angle should be applied to every story. Rather, consider news reportage as a mixture of framing styles and topics. This mixture can have topics and framing styles in different proportions to affect the total toxicity levels of a news organization. In one extreme, we have a news organization that is only reporting on controversial topics with a framing style that is polarizing. This combination, obviously, yields maximal toxicity in audience reactions. The opposite extreme, meaning avoiding controversial topics and reporting on everything with a non-polarizing strategy, would mitigate toxicity. The balance could be found somewhere in between, with a fair coverage of controversial topics using different story framing styles. Thus far, toxicity has not been a factor in editorial decision making, but could it be? This question is worth posing.

The above guidelines highlight the need for an analytical understanding of the toxic behavior of news audiences and seeking ways to mitigate it, within the boundaries and best practices of responsible news reportage. Our findings are not meant to encourage the news media to avoid topics that cause toxicity or blame them for the toxicity. Rather, the findings depict the complex relationship between topics and news audiences. To this end, it is important to note that reading and commenting behavior do not always follow the logic of traditional news standards in deeming whether news is trustworthy or not [77]. News values have shifted dramatically since the advent of online news and online commenting. Bae [78], for example, found that readers who accessed the news via social media had a markedly raised tendency to believe political rumors. In one study, news stories that used sources–traditionally a measure for a story's objectivity–elicited more hostility in the comments sections, while journalist participation in comments raised both the quality of commenting [34].

## Limitations of the study

This research has, naturally, some limitations. First, the research assumes that the topics whose comments are more toxic are also more provocative topics than the topics whose comments are less toxic. However, the existence of toxicity can also have other reasons beyond the video itself, e.g., a toxic exchange between the commentators. In such a case, toxicity is due to not watching the video but due to hostile commentators. News topic, even though important, is not the only factor inciting toxic comments. In contrast, individual posting behaviour is a

determining factor in predicting the prevalence of online hate. For example, Cheng et al. [79] found that, though the baseline rate of online hate was found higher for some topics, user mood and the presence of existing trolling behavior from other users within the context of a discussion doubled users' baseline rates for participating in trolling behavior.

As a social phenomenon, toxic online comments are shaped by many contextual factors [77], including individual psychology and group dynamics. A study by Kaakinen et al. [80] found that online hate increased after the November 2015 Paris terrorist attacks and that wider societal phenomena impact the prevalence of online hate at different times. The complexity of the matter is mirrored in the way research on user comments is dispersed through different disciplines, including journalism studies, communication studies, social psychology, and computer science, making an overarching grasp on the field difficult [77]. These distinct characteristics of online comments underline the fact that users' hateful and toxic responses to certain topics are related to other factors than the topic itself. Future studies should, therefore, aim at synthesizing a conceptual framework of online news toxicity that would include elements of the topic, user-to-user dynamics, and story framing. Based on our findings, these three pillars are essential for understanding toxicity in the news context.

Second, in this research, we make some assumptions that facilitate the analysis but may introduce a degree of error. We assume that the topic of the comment equals the topic of the video where we collected it from. However, it is possible that some comments are off-topic, i.e., not discussing the topic of the video. In such a case, the comment's topic would not match the topic of the video. When interpreting the results, it is useful to consider reader comments to online news content as particular type of text. You et al. [81] describe online comments as "communicative", "parasitic" and "intertextual" (p. 5). Comments share the same platform with the original news item and respond to both the original and to other user comments. Online comments may be generated long after the news item first appeared and may serve user agendas that have very little to do with the original news story.

Third, regarding comment authenticity, it is possible that the sample contains some bot comments. Even though YouTube has filtering mechanisms for bots and the comments that we manually reviewed for this research all seemed real user comments, it is possible that there could be some bot comments. In this regard, we depend on the bot detection applied by YouTube, as bot detection in itself is a complicated subject of research [82]. Overall, we have no reason to believe the above issues would systematically affect a given topic on another topic. Rather, on average, it is likely that toxicity is triggered, to a major degree, by the topic of the video and, on average, the comments deal with the video rather than external stimuli.

Fourth, our analysis omits factors, such as time and user characteristics, that could contribute to toxicity. Unfortunately, as noted in previous research [83], these characteristics are difficult to obtain as social media platforms typically do not expose comment-level user characteristics (e.g., age, gender, country). Here, our focus was on the analysis of topic and toxicity.

Regarding generalizability of the findings, toxic commenting may differ across news organizations and geographical locations. However, the sampled news channel that has a diverse, international audience, reports on a variety of topics from politics to international affairs and has substantial commenting activity among its audience. While these features make it an exemplary case of a modern news channel facing online toxicity, replicating the analysis with content from other channels would be desirable in future work. Moreover, the study was only conducted in English, leaving room for replication in other languages.

## Future research avenues

We identify several fruitful directions for future research. First, future research could investigate how various story framing styles (factual/one-sided/human story, etc.) as well as the linguistic style of news reporting influence the toxic commenting within a topic. Here, we investigated toxicity differences between the topics. As we observe that there is also a variation of toxicity within a topic, future research could explain within-topic variation, for example, by analyzing the impact of linguistic patterns on the average comment toxicity. Other ideas for future research include analyzing data from additional news channels and comparing the results, providing a deeper analysis beyond the included superclass taxonomy, and analyzing the differences between the toxicity levels on YouTube comments and comments in other social media platforms. Finally, research on channel-to-audience interaction is needed, specifically focusing on if and how journalist participation in social media can defuse toxicity.

## Conclusion

Classifying tens of thousands of online videos for news topics and scoring the comments of the videos for toxicity, our empirical analysis reveals an association between online news topics and average comment toxicity. Results highlight the existence of topic-driven toxicity in online news context and provide some suggestions for news channels to potentially alleviate toxicity in their social media channels.

## Supporting information

**S1 File. Python script explaining the data collection.**
(TXT)

**S2 File. Data with full statistical results.**
(XLSX)

**S1 Table. Grouping of data into superclasses.** Note: "religion" was discarded from the analysis because the class contained only 3 videos.
(DOCX)

**S2 Table. Summary of statistical test results.**
(DOCX)

## Author Contributions

**Conceptualization:** Joni Salminen, Soon-gyo Jung, Bernard J. Jansen.

**Data curation:** Sercan Sengün, Soon-gyo Jung.

**Formal analysis:** Sercan Sengün, Juan Corporan.

**Investigation:** Joni Salminen, Soon-gyo Jung.

**Methodology:** Joni Salminen, Juan Corporan, Soon-gyo Jung.

**Project administration:** Joni Salminen, Bernard J. Jansen.

**Resources:** Bernard J. Jansen.

**Supervision:** Joni Salminen, Bernard J. Jansen.

**Writing – original draft:** Joni Salminen, Sercan Sengün, Bernard J. Jansen.

**Writing – review & editing:** Joni Salminen, Bernard J. Jansen.

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
