## [Decision Letter · Decision Letter 0]

5 Nov 2019

PONE-D-19-19498

Topic-driven Toxicity: Exploring the Relationship between Online Toxicity and News Topics

PLOS ONE

Dear Dr. Salminen,

Thank you for submitting your manuscript to PLOS ONE. After careful consideration, we feel that it has merit but does not fully meet PLOS ONE’s publication criteria as it currently stands. Therefore, we invite you to submit a revised version of the manuscript that addresses the points raised during the review process.

Based on the review comments from two experts in this field, all reviewers agree there are merits in this submission and the contributions are significant. At the same time, the reviewers also raised some concerns that are necessary to be addressed for publication. Based on my own reading, I fully agree with the review comments and hope the authors will address the concerns and revise the submission accordingly.

We would appreciate receiving your revised manuscript by Dec 20 2019 11:59PM. To enhance the reproducibility of your results, we recommend that if applicable you deposit your laboratory protocols in protocols.io, where a protocol can be assigned its own identifier (DOI) such that it can be cited independently in the future. For instructions see: http://journals.plos.org/plosone/s/submission-guidelines#loc-laboratory-protocols

We look forward to receiving your revised manuscript.

Kind regards,

Pin-Yu Chen, PhD

Academic Editor

PLOS ONE

Journal Requirements:

"The authors have declared that no competing interests exist.".

We note that one or more of the authors are employed by a commercial company: 'Banco Santa Cruz'.

Additional Editor Comments (if provided):

Based on the review comments from two experts in this field, all reviewers agree there are merits in this submission and the contributions are significant. At the same time, the reviewers also raised some concerns that are necessary to be addressed for publication. Based on my own reading, I fully agree with the review comments and hope the authors will address the concerns and revise the submission accordingly.

Reviewers' comments:

Reviewer's Responses to Questions

**Comments to the Author**

1. Is the manuscript technically sound, and do the data support the conclusions?

Reviewer #1: Yes

Reviewer #2: Yes

2. Has the statistical analysis been performed appropriately and rigorously? 

Reviewer #1: Yes

Reviewer #2: Yes

3. Have the authors made all data underlying the findings in their manuscript fully available?

Reviewer #1: No

Reviewer #2: No

4. Is the manuscript presented in an intelligible fashion and written in standard English?

Reviewer #1: Yes

Reviewer #2: Yes

5. Review Comments to the Author

Reviewer #1: This paper explores the relationship between the topics of news videos posted on Youtube (from the Al Jazeera Media Network), and the toxicity of viewers’ comments. The subject is interesting and timely. The methodology is also interesting, and generally well described. Results show that comment toxicity varies according to the topic presented, which leads the authors to suggest that news comment toxicity is topic-driven (targeting topics) rather than vindictive (targeting people).

While overall I think this is very interesting work, I have some concerns regarding the takeaways (or at least their current presentation) that I believe call for a revision. The paper would also benefit from another pass on typos and grammatical errors.

MAJOR CONCERN

On pages 25, 27, and 28, the authors should absolutely (!!!) remove the names of the persons they quote. Even if they are pseudonyms (which they do not seem to be), the authors should never include any clue that might allow to (re)identify individuals. This is an important breach of research ethics!

REGARDING TAKEAWAYS

In 6.1, the second recommendation seems dangerous to me. Although it follows a trend reported in other work, segmenting news audiences based on their worldview and curating content to comfort those views is precisely how thought echo-chambers are created. I strongly encourage the authors to nuance this point, and to offer some critical reflection on it. For example, while it might be a desirable economic outcome for the outlet, it is very likely not a desirable social outcome.

I am not sure what the very last point in 6.2 adds to the discussion. The authors seem to imply that journalists should not use sources, as they might fuel hostility in comments. This is absurd. More generally, I am a bit concerned by the tension that exists throughout Section 6.2 between adapting journalistic practices to avoid online toxicity in comments, and, as the authors put it: “obviously, [avoiding sensitive topics] is not a good strategy.” While the authors do try to make it explicit that news outlets should not submit to the “tyranny of the audience”, their main arguments and suggestions seem to do exactly that, e.g., by depoliticizing content, or by applying a “humanistic” formula on every story. I recommend the authors rethink and rewrite this section to reduce this tension—especially in light of the first paragraph of 6.3.

OTHER COMMENTS

I think the beginning of Section 3 could be made clearer. What I understand from 3.3 is that the Neural Net is trained on (textual) articles from Al Jazeera’s website, which are tagged with topic labels, and then used to predict the topics of Youtube videos, using their titles and descriptions. Is this correct? This was not clear to me in 3.1.

Further, are all the articles scraped form Al Jazeera’s website in English? The authors seem to imply their data collection is limited to English in 3.4, but do not mention it explicitly.

Is each article on Al Jazeera’s website tagged only once, or can the same article have multiple tags? i.e., is the video-topic-prediction problem a multi-class or multi-label classification problem? The latter would then also complexify the study of the relationship between topic and comment toxicity. That said, it seems from the end of 3.6 that the authors’ Neural Net only outputs one label per video.

I appreciate the authors’ attempts to bring external validation to each step of the data collection and (automatic) annotation process.

I wonder how well connected the comments for a given video are to the topic of that video. Do viewers always stick to what they have watched in the video? Or might they write about other, unrelated topics? Admittedly, the authors do mention this in their discussion, but it would be good to state it earlier in the paper. Also, does each video contain only one topic? (this is related to the multi-label problem mentioned above). Assessing this seems important, since topic toxicity is simply calculated by subsequently averaging the comment toxicity for each video, and the video toxicity for each topic (see 3.8).

I am not sure I understand the procedure for the aggregation of topics into superclasses. Did only one author do this? (Which seems to be the case, but the sentence mentioning this is not clear) Or did several? Also, is there overlap between News Topics and Countries and Regions, or are the topics that are combined into each of these superclasses distinct?

Section 4.2 is very unclear to me. I suggest it be entirely rewritten, possibly condensed, and moved after 4.3 (at least after table 5). What is beta in the equation and in the null hypothesis?

Can table 6 be fit into a single page? Also, the description in Section 4.3 mentions color in the matrix, but I do not see it. It would be good to highlight significant differences in the pairwise comparisons.

In Section 5, the authors mention reading through and (manually?) coding the comments and discussions under the videos. How was the coding done? How many coders were there?

In the subsection on Platform’s power, the authors do not prove there is a causal relationship between Google/Youtube’s description of the news outlet and the comments the they highlight. I understand the commenters are making statements based on the relationship between Al Jazeera and the Qatari government, but claiming this is directly linked to the phrasing of the Youtube tag is a bit of a stretch—which, again, is not empirically proven. What is the proportion of comments that directly target the relationship between the outlet and the government? Does this targeting not occur for the other outlets? Are there instances where Youtube does not tag a video and these types of comments are not present? This seems very one-way focused, and biased towards defending Al Jazeera. I do not think this has its place in the paper. In addition, it seems the wording of the three labels shown in figure 1 simply follows that of the first sentences of the three outlets’ respective Wikipedia pages. It is likely this wording is automatically derived from those sentences. I suggest the authors highly nuance, or even remove this subsection, as well as the discussion in Section 6 on it, as it seems partisan and weakens the rest of the contributions.

In 6.3, one last, important limitation is that the study was only conducted in English.

MINOR

Add white space below each table.

In 6.2, “it is becoming increasingly difficult for news media to remain [un]biased…”

Also in 6.2, “—intentionally or [un]intentionally—”

Reviewer #2: The theme of the paper is interesting, overall paper is well written and well organized. Moreover, the analysis have been rigorously performed and results are presented appropriately. I have only a few comments that are given below.

In this paper, a third party service is used for toxicity quantification that was unable to compute toxicity on 21.8% of the comments that were likely to be not written in English. To tackle this, language detection of the users’ comments can be performed earlier to toxicity analysis.

Please cite a reference to support the argument regarding manual tagging by Al Jazeera’s journalists and editors for topical keywords, if any.

A few state of the art machine learning (ML) methods can be used for performance evaluation purposes, e.g., it would be interesting to compare the performance of traditional ML methods like decision trees classifier with that of the feed forward neural network. Moreover, performance evaluation can also help in selecting a suitable ML method for analysis.

6. PLOS authors have the option to publish the peer review history of their article (what does this mean?). If published, this will include your full peer review and any attached files.

Reviewer #1: No

Reviewer #2: No

---

## [Author Response · Author response to Decision Letter 0]

10 Dec 2019

Reviewer #1: This paper explores the relationship between the topics of news videos posted on Youtube (from the Al Jazeera Media Network), and the toxicity of viewers’ comments. The subject is interesting and timely. The methodology is also interesting, and generally well described. Results show that comment toxicity varies according to the topic presented, which leads the authors to suggest that news comment toxicity is topic-driven (targeting topics) rather than vindictive (targeting people). 

>>Thank you for seeing the value in our research!

While overall I think this is very interesting work, I have some concerns regarding the takeaways (or at least their current presentation) that I believe call for a revision. The paper would also benefit from another pass on typos and grammatical errors. We thank the reviewer for this comment.

>>The manuscript has now undergone another round of proof-reading. Changes made are highlighted in yellow color in the manuscript.

On pages 25, 27, and 28, the authors should absolutely (!!!) remove the names of the persons they quote. Even if they are pseudonyms (which they do not seem to be), the authors should never include any clue that might allow to (re)identify individuals. This is an important breach of research ethics!

>>Thanks for noticing - these have been removed.

In 6.1, the second recommendation seems dangerous to me. Although it follows a trend reported in other work, segmenting news audiences based on their worldview and curating content to comfort those views is precisely how thought echo-chambers are created. I strongly encourage the authors to nuance this point, and to offer some critical reflection on it. For example, while it might be a desirable economic outcome for the outlet, it is very likely not a desirable social outcome.

>>We have revised Section 6.1 by removing the recommendations the reviewer considers dangerous. We ask the reviewer to take a look and provide suggestions for additional changes if needed.

I am not sure what the very last point in 6.2 adds to the discussion. The authors seem to imply that journalists should not use sources, as they might fuel hostility in comments. This is absurd. More generally, I am a bit concerned by the tension that exists throughout Section 6.2 between adapting journalistic practices to avoid online toxicity in comments, and, as the authors put it: “obviously, [avoiding sensitive topics] is not a good strategy.” While the authors do try to make it explicit that news outlets should not submit to the “tyranny of the audience”, their main arguments and suggestions seem to do exactly that, e.g., by depoliticizing content, or by applying a “humanistic” formula on every story. I recommend the authors rethink and rewrite this section to reduce this tension—especially in light of the first paragraph of 6.3.

>>The reviewer’s comments are insightful.

>>However, this is not an easy problem to address.

>>Actually, it is a useful recommendation to frame stories on sensitive topics using human stories, as our findings suggest this can alleviate the conflict among social media audiences, if this is an organizational objective. Overall, there are very few practical recommendations for journalists in prior literature on how to address toxicity, or even consider it as a factor they could have control over. Thus, providing suggestions about story framing is useful and, naturally, editors and journalists can freely choose if they want to make use of these suggestions. We would love to hear some constructive ideas from the reviewer on this concerning other suggestions for journalistic practice.

>>Regarding the aspect of tension, we do agree to some extent with the reviewer’s comment. The tension, however, is not of our making, but it follows from the state of the matters in the real world: if topics affect toxicity, then toxicity can be reduced by avoiding topics. This is just logical fact-based argument, not a recommendation of how things “should be” from a moral perspective.

>>How things should be, in an ideal world, is that news channels and journalists would be perceived as objective by the audience, without agendas and biases. If opposite perceptions take place among the audience, as we increasingly see from social media commenting, the result will be more anti-media sentiments, alternative news, and judging traditional news media as “fake news” that only report on certain topics with certain sentiments. 

>>So, there *is* a serious conflict taking place in the real world between news audiences and news reporting. To be fair, this tension should not be hidden from the discussion of our results.

>>As a side note, “depoliticizing” content does not mean avoiding political topics as the reviewer might imply. It means defusing a controversial topic by using a framing style that is aimed at defusing toxicity while maintaining high journalistic standards. We have now clarified this in p. 29.

>>To illustrate, consider a binary choice: given the journalist knows Topic A is controversial, does the story framing strategy aim to (a) exacerbate controversy or (b) alleviate controversy. This strategic choice, we argue, is important for the toxicity outcome.

>>Also note that we do not make the argument that human story angle should be applied to “every story”. Rather, consider news reportage as a mixture of framing styles and topics. This mixture can have topics and framing styles in different proportions to affect the total toxicity levels of a news organization. In one extreme, we have a news organization that is only reporting on controversial topics with a framing style that is polarizing. This combination, obviously, yields maximal toxicity in audience reactions. The opposite extreme, meaning avoiding controversial topics and reporting on everything with a non-polarizing strategy, would mitigate toxicity. The balance could be found somewhere in between, with a fair coverage of controversial topics using different story framing styles. Thus far, toxicity has not been a factor in editorial decision making, but could it be? This question is worth posing.

>>From a philosophical point of view, one can consider this as a choice of worldviews: there can be a choice of worldview that states “toxicity is out there, it is external, and I (as a media agency) have no control over it”. Or, there can be an alternative view: “toxicity is out there, but I (as a media agency) can find ways to mitigate it with my own actions.” According to our experiences with journalists in this media organization, most of them subscribe to the first worldview ---- but, that does not mean that we, as researchers, could not increase their awareness over the potential ways toxicity could be defused. From an ethical point of view, the opposite seems like an imperative; and, if anything, we are proposing too few framing styles here. Future research should pursue these styles in much greater detail.

>>Again, we welcome any suggestions on how to improve the discussion – but we cannot shy away from the tension of media being an actor in online toxicity. Topics and story framing are associated with toxicity – where to go from there is the million-dollar question.

I think the beginning of Section 3 could be made clearer. What I understand from 3.3 is that the Neural Net is trained on (textual) articles from Al Jazeera’s website, which are tagged with topic labels, and then used to predict the topics of Youtube videos, using their titles and descriptions. Is this correct? This was not clear to me in 3.1.

>>Correct. We have clarified this in the following terms (p. 10):

“In other words, the FFNN is trained on textual articles from Al Jazeera’s website, which are tagged with topic labels, and then used to predict the topics of YouTube videos, using their titles and descriptions.”

Further, are all the articles scraped form Al Jazeera’s website in English? The authors seem to imply their data collection is limited to English in 3.4, but do not mention it explicitly.

>>Yes, in English. We have clarified this in the text (see p. 11).

Is each article on Al Jazeera’s website tagged only once, or can the same article have multiple tags? i.e., is the video-topic-prediction problem a multi-class or multi-label classification problem? The latter would then also complexify the study of the relationship between topic and comment toxicity. That said, it seems from the end of 3.6 that the authors’ Neural Net only outputs one label per video.

>>Good question. It is a multilabel classification problem, as one news article can contain many labels. We have modeled the problem as such, and now clarify this in p. 13:

>>“From a technical point of view, this is a multilabel classification problem, as one news article is typically labeled for several topics.”

>>Regarding the end of 3.6, the output of the classifier is confidence value for each news story and each topic (of which, the topics are the ones exceeding a set threshold value for the confidence, in our case the commonly used value of 0.5). For parsimony, we choose the topic with the highest confidence value for the statistical analysis, i.e., a story has only one dominant topic in the statistical analysis. Using all topics would make the statistical comparison exceedingly complex. We now mention this point explicitly in p. 14.

>>[Reference to other comment: **]

I appreciate the authors’ attempts to bring external validation to each step of the data collection and (automatic) annotation process.

>>Thank you!

I wonder how well connected the comments for a given video are to the topic of that video. Do viewers always stick to what they have watched in the video? Or might they write about other, unrelated topics? Admittedly, the authors do mention this in their discussion, but it would be good to state it earlier in the paper. Also, does each video contain only one topic? (this is related to the multi-label problem mentioned above). Assessing this seems important, since topic toxicity is simply calculated by subsequently averaging the comment toxicity for each video, and the video toxicity for each topic (see 3.8).

>>Regarding the multi-label point, this was mentioned in relation to the previous comment (see reference to other comment: **).

>>The “off-topic” factor is treated as random noise in our analysis, in the sense that we assume each comment for each video to be equally likely to have off-topic elements. This assumption, although somewhat naïve, is required because the alternative would be to explicitly model the topic structure of the comments. Doing so could possibly break the association between the comments discussing the video (with a known topic) unless the classification error would be much smaller than that of the video topic classification. It is likely that, given the performance of short-text classification, attempting a comment-specific classification would introduce another source of error, which in our opinion could be higher than that of leaving the off-topic comments in place and considering them as random noise. Most certainly, it would add to the complexity of the research design that, we feel, is already complex enough with several steps of analysis.

>>So, in conclusion, while we agree with the reviewer that in the ideal case each comment would be evaluated for off-topic content, in reality, we must maintain this being a limitation of our particular study.

I am not sure I understand the procedure for the aggregation of topics into superclasses. Did only one author do this? (Which seems to be the case, but the sentence mentioning this is not clear) Or did several? Also, is there overlap between News Topics and Countries and Regions, or are the topics that are combined into each of these superclasses distinct?

>>The aggregation was done by one of the researchers and verified by another researcher, now clarified in p. 17.

>>Regarding news topics and countries and regions, the taxonomy treats these as equal classes (not parallel or hierarchical). This means that if the classifier gives “Middle East” as the highest and “Science & Technology” as second highest, in the statistical analysis we would use “Middle East” as the class of the story. (Also see reference to other comment: **.)

Section 4.2 is very unclear to me. I suggest it be entirely rewritten, possibly condensed, and moved after 4.3 (at least after table 5). What is beta in the equation and in the null hypothesis?

>>We have revised this section and moved it after Table 5 (now Table 4). Please refer to p. 18.

>>Beta is the estimated regression coefficient (mentioned now in p. 18).

>>If anything else is unclear, please ask. Happy to clarify!

Can table 6 be fit into a single page? Also, the description in Section 4.3 mentions color in the matrix, but I do not see it. It would be good to highlight significant differences in the pairwise comparisons.

>>To address this comment, we have replaced the table with Figure 1 (p. 20). This figure communicates the statistically significant differences better, while also demonstrating the variation in some of the multiple comparison test results. We hope the reviewer finds this solution satisfactory.

In Section 5, the authors mention reading through and (manually?) coding the comments and discussions under the videos. How was the coding done? How many coders were there?

>>Page 21 now explains the method for qualitative analysis in greater detail. Essentially, it was carried out as a collaborative effort between two researchers.

In the subsection on Platform’s power, the authors do not prove there is a causal relationship between Google/Youtube’s description of the news outlet and the comments the they highlight. I understand the commenters are making statements based on the relationship between Al Jazeera and the Qatari government, but claiming this is directly linked to the phrasing of the Youtube tag is a bit of a stretch—which, again, is not empirically proven. What is the proportion of comments that directly target the relationship between the outlet and the government? Does this targeting not occur for the other outlets? Are there instances where Youtube does not tag a video and these types of comments are not present? This seems very one-way focused, and biased towards defending Al Jazeera. I do not think this has its place in the paper. In addition, it seems the wording of the three labels shown in figure 1 simply follows that of the first sentences of the three outlets’ respective Wikipedia pages. It is likely this wording is automatically derived from those sentences. I suggest the authors highly nuance, or even remove this subsection, as well as the discussion in Section 6 on it, as it seems partisan and weakens the rest of the contributions.

>>We agree with the reviewer and have removed this subsection from the results and discussion.

In 6.3, one last, important limitation is that the study was only conducted in English.

>>Added – see p. 33:

>>“Moreover, the study was only conducted in English, leaving room for replication in other languages.”

Add white space below each table.

>>Done.

In 6.2, “it is becoming increasingly difficult for news media to remain [un]biased…” 

>>We’ve corrected this sentence as follows (p. 5):

>>“it is becoming increasingly difficult for news media to provide facts without seen as a manipulator or stakeholder in the debate itself.”

Also in 6.2, “—intentionally or [un]intentionally—”

>>Fixed (see p 29).

Reviewer #2: The theme of the paper is interesting, overall paper is well written and well organized. Moreover, the analysis have been rigorously performed and results are presented appropriately. 

>>Thank you!

In this paper, a third party service is used for toxicity quantification that was unable to compute toxicity on 21.8% of the comments that were likely to be not written in English. To tackle this, language detection of the users’ comments can be performed earlier to toxicity analysis.

>>Good point! While this could have been done separately using a language detection approach, since Perspective API outputs an error for the non-English comments, a separate analysis was not seen necessary. 

Please cite a reference to support the argument regarding manual tagging by Al Jazeera’s journalists and editors for topical keywords, if any.

>>Good suggestion! We have justified the use of journalists (=domain experts) for training data annotation as follows (p. 14):

>>“The importance of using domain experts for data annotation is widely acknowledged in ma-chine learning literature [54,55]. Generally, expert taxonomies are considered as gold stand-ards for classification [56].”

A few state of the art machine learning (ML) methods can be used for performance evaluation purposes, e.g., it would be interesting to compare the performance of traditional ML methods like decision trees classifier with that of the feed forward neural network. Moreover, performance evaluation can also help in selecting a suitable ML method for analysis.

>>Yes, we actually did compare the neural network (NN) to Random Forest (RF) which is a tree-based method as suggested by the reviewer. The overall performance was better for NN (F1=0.700) relative to RF (F1=0.458). This is now explained in p. 14:

>>“It also clearly outperforms a Random Forest (RF) model that was tested as a baseline model (F1RF=0.458).”

>>We acknowledge there are other algorithms to test as well, such as LightGBM and XGBoost that have obtained high performance in text classification. However, for the purposes of this study, we consider the performance obtained with our FFNN as adequate. The contribution of the paper is not technical but rather focused on the analysis of the relationship between toxicity and news topics.

Upon re-submitting your revised manuscript, please upload your study’s minimal underlying data set as either Supporting Information files or to a stable, public repository

>>The data and full results (using various correction methods) have been uploaded as a Supporting Information file.

---

## [Decision Letter · Decision Letter 1]

23 Jan 2020

Topic-driven Toxicity: Exploring the Relationship between Online Toxicity and News Topics

PONE-D-19-19498R1

Dear Dr. Salminen,

We are pleased to inform you that your manuscript has been judged scientifically suitable for publication and will be formally accepted for publication once it complies with all outstanding technical requirements.

With kind regards,

Pin-Yu Chen, PhD

Academic Editor

PLOS ONE

Additional Editor Comments (optional):

The revised version has addressed the reviewers' concerns from the previous round. I thank the authors and the reviewers for making great efforts in improving this submission. I recommend to accept this version for publication as is.

Reviewers' comments:

Reviewer's Responses to Questions

**Comments to the Author**

1. If the authors have adequately addressed your comments raised in a previous round of review and you feel that this manuscript is now acceptable for publication, you may indicate that here to bypass the “Comments to the Author” section, enter your conflict of interest statement in the “Confidential to Editor” section, and submit your "Accept" recommendation.

Reviewer #2: All comments have been addressed

2. Is the manuscript technically sound, and do the data support the conclusions?

Reviewer #2: Yes

3. Has the statistical analysis been performed appropriately and rigorously? 

Reviewer #2: Yes

4. Have the authors made all data underlying the findings in their manuscript fully available?

Reviewer #2: Yes

5. Is the manuscript presented in an intelligible fashion and written in standard English?

Reviewer #2: Yes

6. Review Comments to the Author

Reviewer #2: (No Response)

7. PLOS authors have the option to publish the peer review history of their article (what does this mean?). If published, this will include your full peer review and any attached files.

Reviewer #2: No

---

## [Editor Report · Acceptance letter]

12 Feb 2020

PONE-D-19-19498R1 

Topic-driven Toxicity: Exploring the Relationship between Online Toxicity and News Topics 

Dear Dr. Salminen:

I am pleased to inform you that your manuscript has been deemed suitable for publication in PLOS ONE. Congratulations! Your manuscript is now with our production department. 

With kind regards,

on behalf of

Dr. Pin-Yu Chen 

Academic Editor

PLOS ONE